# Design of a Gabor Filter-Based Image Denoising Hardware Model

**Virodhi Dakshayani** [1], **Govinda Rao Locharla** [1,*], **Paweł Pławiak** [2,3,*], **Venkataramana Datti** [1] **and Chiranjeevi Karri** [1]

1 Department of Electronics and Communications Engineering, GMR Institute of Technology Rajam, Srikakulam 532127, AP, India; shiny.v0973@gmail.com (V.D.); venkataramana.d@gmrit.edu.in (V.D.); chiranjeevi.k@gmrit.edu.in (C.K.)
2 Department of Computer Science, Faculty of Computer Science and Telecommunications, Cracow University of Technology, 31-155 Krakow, Poland
3 Institute of Theoretical and Applied Informatics, Polish Academy of Sciences, 44-100 Gliwice, Poland
* Correspondence: govindarao.l@gmrit.edu.in (G.R.L.); plawiak@pk.edu.pl or plawiak@iitis.pl (P.P.)

**Abstract:** The intervention of noise into images during data acquisition and transmission is inevitable. Hence, the denoising of such affected images is essential in order to have effective image analysis where it needs image filtering. The Gabor filter is widely adapted in various image processing applications for feature extraction, texture analysis, pattern analysis, etc. The Gabor-based filtering technique adopted in work is aimed for image filtering in order to extract edges. The design of a low-power portable system deploys hardware accelerators to achieve high performance per watt in feature extraction and edge detection. In this paper, an image denoising hardware accelerator model is mapped from the Gabor filter function. Moreover, hardware models for realizing various parameters involved in the Gabor function are also presented. A MATLAB model for the proposed denoising hardware accelerator is simulated and performance is measured in terms of the peak-signal-to-noise ratio, mean square error, histograms and compared with algorithm level performance reported in the literature. It is observed that the proposed hardware architecture model showed better performance compared to the mathematical models reported in the literature. However, the key limitation is the degradation of hardware performance due to a truncation or rounding of the sample's word length.

**Keywords:** gabor filter; hardware accelerator; edge detection; peak-signal to noise ratio; mean square error; histogram

## 1. Introduction

In biomedical systems, the invasion of noise into imaging such as Ultrasound, CT scan, MRI scan, etc., is inevitable at the stages of image capture, data acquisition and transmission. Such applications need denoising to have an effective analysis of the image and the diagnosis. Image processing is performed to obtain an enhanced image or a set of characteristics or features of the input image. Images are characterized by various parameters such as object contour, orientation, size and color. Image processing systems play a crucial role in applications such as the Internet of Things (IoT) and deep learning [1,2]. The most interesting application of image processing is image filtering. Certain features can be emphasized or removed by filtering an image with the help of filtering techniques. The operations dealing with image processing are executed with filtering process such as smoothing, sharpening and edge enhancement, which improves the quality of an image by rectifying blurred images, reducing noise and edge detection. In most image processing applications, the Gabor function is widely adopted for image filtering [3]. The Gabor filter is a linear filter that performs convolution to eliminate noise.

The Gabor filter (GF) minimizes the joint uncertainty for frequency and spatial orientation [3]. The Gabor filter is named after Dennis Gabor, which basically determines



the frequency components of the image with a particular orientation in a localized region around the point or region of evaluation. Frequency and orientation expressions of the Gabor filter are identical to that of a human visual system, and it is specifically applicable for texture characterization and differentiation. The popularity of the Gabor filter is involved in many applications such as texture classification, object recognition [4], iris recognition techniques, edge and corner detection [5], visual search (VS), gait analysis, image segmentation, speech recognition and also in multimedia processing applications, where the performance of the mammalian visual cortex cells is well illustrated mostly due to the family of filters [6].

During the last few decades, the obtainability of new electronic devices underwent crucial development in many fields. The Gabor filter is also used in medical applications, e.g., magnetic resonance (MR), X-ray computed tomography (CT) scan. In several papers, The Gabor filter is proved and shown that it has an ability to provide a better performance when it is instantly used for image enhancement and feature extraction, which involves the invariance features of rotation, scaling and translation.

### 1.1. Related Works

Gabor-based image processing gained a significant research focus during the recent decades after the combined time and frequency domain representation of the signals reported by D. Gabor in [3]. In [7], J.-K. Kamarainen et al. presented a methodology for the identification of two-dimensional (2D) fixed entities such as orientation, illumination, scale and translation. In [5], R. Mehrotra et al. addressed a design criterion for one-dimensional (1D) Gabor filter-based edge detector and presented the performance analysis for a two-dimensional (2D) GF-based edge detector, which was also discussed and holds true. G. D. Licciardo et al. [8] reported a state-of-the-art architecture for the processing of high dynamic range (HDR) version images from a low dynamic range image (multiresolution), which can process the images of various resolutions obtained from an image sensor. In [9], Lubna, M. F. Khan et al. presented a comparison of various edge detection filters with the use of variant pre-processing techniques. In [10], G. Humpire-Mamani et al. developed a novel k-Gabor method by using the clustering algorithm.

From the literature, it was witnessed that HW solutions are preferred in high speed applications related to data transfer and data pre-processing for multimedia [3,11–16]. G. D. Licciardo et al. presented a processor implementation for the expansion of full-HD LDR images to HDR to elaborate the streaming images to obtain real-time performances [17]. In [6], G. D. Licciardo et al. reported a 2D Convolution based filtering of images and videos by using hardware architectures for visual applications. Here, both the arithmetic and memory modules were also developed to reduce the mapped resources and to improve the throughput. An HW accelerator for image processing applications is developed by C. Cappetta et al. [18]. G. D. Licciardo et al. [19] designed a hardware architecture for Gabor filter-based edge detection for visual searches. In [20], G. D. Licciardo et al. proposed three different designs of GF for medical imaging applications with trade-offs between various specifications. These designs have been targeted toward FPGA and synthesized relative to CMOS standard cell implementation. In [21], E. kayalvizhi et al. proposed an algorithm for a low-power architecture of 2D Gabor filter. Here, the performance was analyzed by filtering with various noises and by comparing with median filters. In [22], K. R. Namuduri et al. explored a technique with the help of Canny's measures to analyze the performance of Gabor edge detection. Ke Wang et al. introduced a method for detecting edges by using FFT [23]. A. H. A. Razak and R. H. Taharim demonstrated an approach to improve the fingerprint image by the application of Gabor filters and reported its hardware implementation using Verilog HDL in [24].

Carmine Cappetta et al. presented a novel Gabor-based hardware accelerator design for input data filtering in [11]. This design targets to a Xilinx Virtex 7 board to obtain a minimum operating clock period. In [25], Amira Hadj Fredj and Jihene Malek implemented a high-level synthesis on a Zedboard platform, in this design, a DMA peripheral has been

chosen to optimize hardware acceleration. The evaluation of the filter has been performed with the use of various medical images. In [26], Anirban Sengupta et al. presented a novel low power multi-model $3 \times 3$ and $5 \times 5$ generic filter hardware accelerators for applications such as blurring, embossment, sharpening and edge detection by changing the control input. Goyal, Bhawna and Ayush Dogra et al. presented the reports on evaluation, comparison and classification of different image denoising methods; those are partitioned into five domains. Here, the fundamental aim of image denoising is edge preservation by eliminating noisy pixels [27]. In [28], R. Harikumar et al. focused on Singular value Decomposition (SVD), RBF and Elman networks for the categorization of epilepsy risk levels acquired from code converters using EEG signals parameters, which are produced by morphological operators.

*1.2. Contributions of This Paper*

In this paper, a novel architecture for the Gabor filter-based image denoising hardware accelerator mapped from the Gabor filter math is presented. Moreover, hardware models for realizing the fractional values of various parameters involved in the Gabor function are also presented. The proposed Gabor filter-based image denoising hardware accelerator is designed for Edge detection in image processing applications. The MATLAB model for the proposed denoising hardware accelerator is simulated and the performance is measured in terms of the peak-signal-to-noise ratio, mean square error and compared to the benchmark performance reported in the literature. It is observed that the proposed hardware architecture design is consistent with the performance of the mathematical models reported in the literature.

The paper is organized as follows. The theoretical background of the work is focused on in Section 2. The proposed framework is presented in Section 3. In Section 4, the proposed architecture of the Gabor filter is presented. The performance comparison and result analysis are discussed in Section 5. Finally, Section 6 concludes the work.

**2. Theoretical Background of Gabor Filter**

The Gabor Filter is a linear filter for which its impulse response is determined by a sinusoidal wave (i.e., a level of 2D Gabor filters) convoluted by a Gaussian function. As per the convolution theorem, the Fourier transform (FT) of a Gabor's filter impulse response is generated by convoluting the Fourier transform of a harmonic function (sinusoidal function) and the Fourier transform of a Gaussian function.

$$FT(Gabor) = FT(Harmonic) \times FT(Gaussian) \tag{1}$$

The Gabor filter is a set of band pass filters that compute within a frequency range to accept or reject the computations performed on the filter. By modulating the Gaussian kernel function with a sinusoidal wave, edges, textures and feature extractions can be found easily with the use of a Gabor filter.

The general function for a 2D Gabor filter in the spatial domain is defined by the following expression.

$$\Psi(x, y; f_0, \theta) = \frac{f_0^2}{\pi \gamma \eta} e^{-\left\{ \frac{f_0^2}{\gamma^2} \cdot x'^2 + \frac{f_0^2}{\gamma^2} \cdot y'^2 \right\}} \tag{2}$$

From Equation (2), the geometric coordinates of the Gabor kernel are $x$ and $y$; $f_0$ is the central frequency at a certain point; $\theta$ is the rotation angle for a specific orientation of both the major axes of Gaussian function and the plane wave; and the spatial aspect ratio is denoted by $\gamma$, which specifies the ellipticity of the Gaussian across the major axis.

The Gabor filter represents the orthogonal direction with the complex component. The complex component consists of both even and odd functions. The even function represents the real part of the Gabor function, while the odd one represents the imaginary part.

From Equation (2), the real part of Gabor function and imaginary part are rearranged as (3):

$$G(x,y;f_0,\theta) = \frac{f_0^2}{\pi\gamma\eta}e^{-\left(\frac{f_0^2}{\gamma^2}\cdot x'^2 + \frac{f_0^2}{\gamma^2}\cdot y'^2\right)}\cdot cos(2\pi f_0 x')$$
$$+jsin(2\pi f_0 x') \tag{3}$$

where

$$x' = xcos\theta + ysin\theta$$
$$y' = -xsin\theta + ycos\theta \tag{4}$$

Equation (4) provides the highest response of the filter by controlling the Gabor's filter central frequency $(f_0)$. The Gabor filter uses many parameters that can be tuned or changed to extract useful information. These parameters are described below.

### 2.1. Gabor Kernel Size (k Size)

The size of the kernel of the Gabor filter is denoted by $k$ size. The kernel size can either be a smaller or larger size; by the use of a smaller sized kernel, the images appear to find some particular features where kernels with larger size handles larger objects. The kernel size can be of any size such as $3 \times 3, 7 \times 7, 9 \times 9, 15 \times 15 \ldots 31 \times 31, 33 \times 33$ and so on. If the kernel size exceeds 31 (i.e., $k$ size = 31), then no change is seen in the filtered image by showing some invariance. Large images obtain different outputs; moreover, at a certain point, large kernel sizes show no variance in the filtered image (it almost matches with the input image); this is known as the Gaussian blurring effect.

### 2.2. Orientation (θ)

Theta $\theta$ rotates at different angles at a particular direction of the filter within the sinusoidal wave. $\theta$ varies from $0°$–$360°$. For every transverse of this parameter, it provides different features and extractions from the image. Based on this orientation, the filter will find features. However, at some angles, certain features may not be seen or found; thus, those can be combined with other different extractions to expose more features.

### 2.3. Central Frequency $(f_0)$

The range of the Gabor filter's central spatial frequency will always be in between [0, 0.5] cycle/pixels. According to the Nyquist theorem, the maximum frequency is considered as $(f_n)$ = 0.5 cycle/pixels. For central frequencies less than 0.1 cycle/pixels, it causes some breakdowns in characteristic precision due to improper clearance in the filtered image. Likewise, frequencies with more than 0.2 cycle/pixels and filters with narrow bandwidths are required for undesired excess sampling of the kernel with reference to the grid persistence of the input image. In [20], it has been shown that by considering central frequencies of $(f_0)$ = 0.2 cycle/pixels, the blurring effect is minimized and also compensates the above constraints.

### 2.4. Gaussian Sharpness (γ and η)

Gamma $\gamma$ is the sharpness through the longer axis of the Gaussian, which controls the filter height, whereas $\eta$ is the sharpness of the Gaussian across the shorter axis. As the value of $\gamma$ becomes lower, the filter becomes closer to the pixel size, while a larger value will correlate the filter to the image's size. The blurring effect decreases for higher values, which resembles the kernel behavior with small heights of the filter. This is nothing, but more information can be gathered when the kernel moves through the image.

## 3. Methodology

The Gabor filtering is a popular method used for texture feature analysis and it can be realized by performing convolution on the Gaussian function with trigonometric functions in a two-dimensional space. By selecting the appropriate Gabor function, different scales and directional features can be recognized from the input image. This enables usage of the Gabor filter in image denoising and edge detection applications. In this section, representation of Gabor function and its application in image denoising and edge detection are presented.

Considering the statistical expressions of GF, the 2D-Gabor filter can be represented as in Equation (1).

$$G(x,y) = S(x,y) * W(x,y) \tag{5}$$

The Equation (5) represents the convolution between $S(x,y)$ and $W(x,y)$ where, $S(x,y)$ and $W(x,y)$ are expressed as in Equations (6) and (7).

$$S(x,y) = e^{-j2\pi(u_0 x + v_0 y)} \tag{6}$$

$$W(x,y) = \frac{1}{\sqrt{2\pi\sigma}} e^{-\frac{1}{2}\left(\frac{x^2}{\sigma_x^2} + \frac{y^2}{\sigma_y^2}\right)} \tag{7}$$

Therefore, the Equations (5) can be expressed as in Equation (8)

$$G(x,y) = e^{-\frac{1}{2}\left(\frac{x^2}{\sigma_x^2} + \frac{y^2}{\sigma_y^2}\right)} \cdot e^{-j2\pi(u_0 x + v_0 y)} \tag{8}$$

Equation (8) can be further simplified as in Equation (9)

$$G(x,y;\sigma,\psi,\gamma,\lambda,\theta) = e^{-\frac{1}{2}\left(\frac{x'^2 + \gamma^2 y'^2}{\sigma^2}\right)} e^{i\left(2\pi\frac{x'}{\lambda} + \psi\right)} \tag{9}$$

From Equation (9),

$$\Re\{G(x,y;\sigma,\psi,\gamma,\lambda,\theta)\} = e^{-\frac{1}{2}\left(\frac{x'^2 + \gamma^2 y'^2}{\sigma^2}\right)} \cos\left(2\pi\frac{x'}{\lambda} + \psi\right) \tag{10}$$

and

$$\Im\{G(x,y;\sigma,\psi,\gamma,\lambda,\theta)\} = e^{-\frac{1}{2}\left(\frac{x'^2 + \gamma^2 y'^2}{\sigma^2}\right)} \sin\left(2\pi\frac{x'}{\lambda} + \psi\right) \tag{11}$$

The significance of $\sigma, \psi, \gamma, \lambda, \theta$, involved in the Gabor function formula can be defined as follows. The parameter Lambda ($\lambda$) denotes the wavelength. The value of $\lambda$ can be more than or equal to 2 but not more than $\frac{1}{5}$th of input image dimensions. The image becomes more clearer when $\lambda$ reaches a larger value.

$\psi$ is the phase parameter of a Gabor kernel function. The kernel bands move either to the right side or the left side as Gabor filters are utilized as a bank of filters to find the number of filters in the bank. Its value is determined in the range of $[-180°$ to $180°]$; among them, the values identical to $0°$ and $180°$ are equal to origin and the equations $-90°$ to $90°$ are symmetrical about the center.

The ratio of $\frac{\sigma}{\lambda}$ represents the bandwidth of the half-response spatial frequency of the filter that differs between upper and lower frequencies. In Equations (12) and (13), $\sigma$ indicates the standard deviation of the Gaussian factor of the Gabor filter.

$$Bandwidth\ (b) = log_2 \frac{\frac{\sigma}{\lambda}\cdot\pi + \sqrt{\frac{ln2}{2}}}{\frac{\sigma}{\lambda}\cdot\pi - \sqrt{\frac{ln2}{2}}} \tag{12}$$

$$\frac{\sigma}{\lambda} = \frac{1}{\pi} \cdot \sqrt{\frac{ln2}{2}} \cdot \frac{2^b + 1}{2^b - 1} \tag{13}$$

The $\sigma$ value varies with bandwidth $b$, and it cant be set instantaneously. The standard deviation can be expressed in terms of the wavelength as $\sigma = 0.56\lambda$.

### 3.1. Gabor Filter Based Image Denoising

The block diagram of the proposed Gabor filter based image denoising architecture is shown in Figure 1.

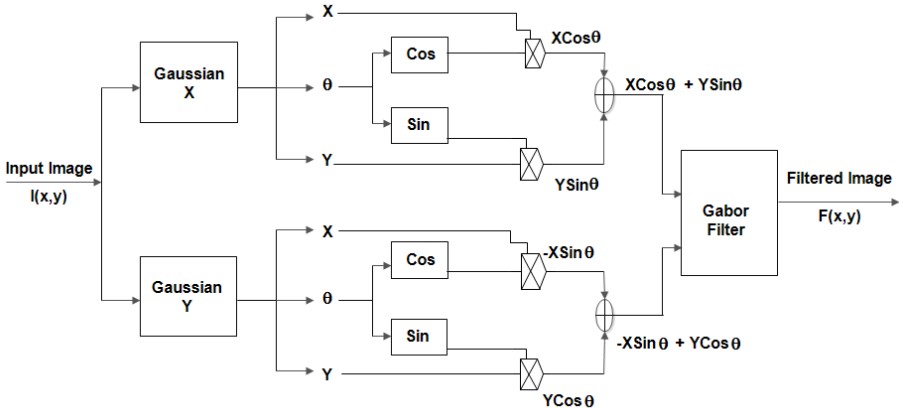

**Figure 1.** Block diagram of the proposed Image Denoising using the Gabor Filter.

The RGB image is converted to a grey-scale image at pre-processing stage where a pixel size of 8-bits is chosen to have the range of quantization levels from 0 to 255. Here, each pixel is a shade of grey between black and white. In this work, $256 \times 256$ pixel size is considered for the input. Figure 2 shows the grey image for the given input test image. The Grey image is then transformed into a binary image. The intensity values in a binarized image for each pixel can either be black or white: 0 for black and 1 for white. These types of images are more efficient for storage and are used in many applications such as fingerprints, texts, etc. The image obtained here is $I(x, y)$ where the spatial coordinates $x$ and $y$ will later undergo a filtering process with kernel functions. Input image $I(x, y)$ is convolved with Gabor kernel coefficient $gb(x, y)$ to form a filtered image $F(x, y)$. The Gabor kernel filters the input image for certain information.

The values considered various parameters involved in Gabor filter for image denoising are considered as follows: $\lambda = 3.5$, $\theta = 0$ or $\pi/4$ or $\pi/2$ or $3\pi/4$, $\psi = 0$, $\sigma = 2.8$, $\gamma = 0.3$, and $\pi = 180$. These values are applied in the kernel function for the respective parameter. The image features gained during the filtering process with the Gabor kernel function are constant in terms of the spinning of the image and scaling in frequency [3]. It offers less memory usage, low power consumption, improved noise reduction and improved data transfer rate. The Gabor filter is an effective linear filter that requires intensive calculations in the filtering of an image. Convolution process takes place between input images and the filter kernel to form the filtered image.

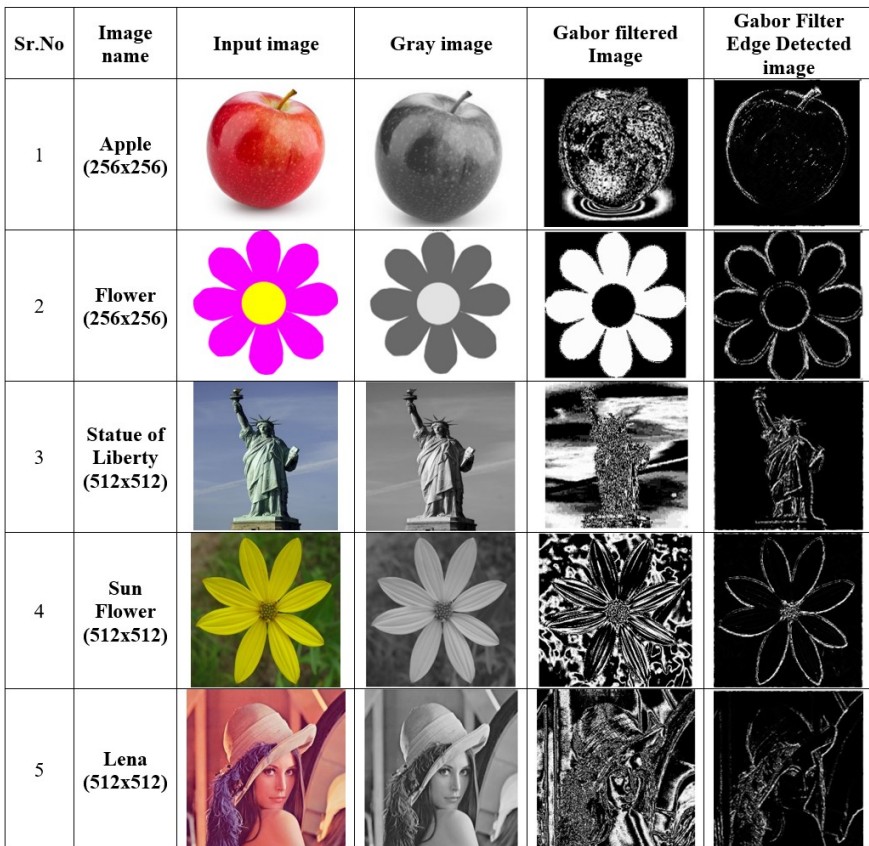

**Figure 2.** The Gabor filtered and edge detection outputs for various input images.

### 3.2. Gabor Filter Based Edge Detection Method

The steps involved in Gabor filter based edge detection process are illustrated in Figure 3. In edge detection applications, the odd function of the Gabor filter is applied for extracting the edge particulars of the image and for achieving good robustness results relative to noise. Edge information in different directions of the image is detected by constructing the filter banks with various scales and directions with the parameters described, then the non-edge points are suppressed, and the edge information is combined in different directions to obtain complete edge information. At the end, all edges of the image are concatenated to obtain edge information.

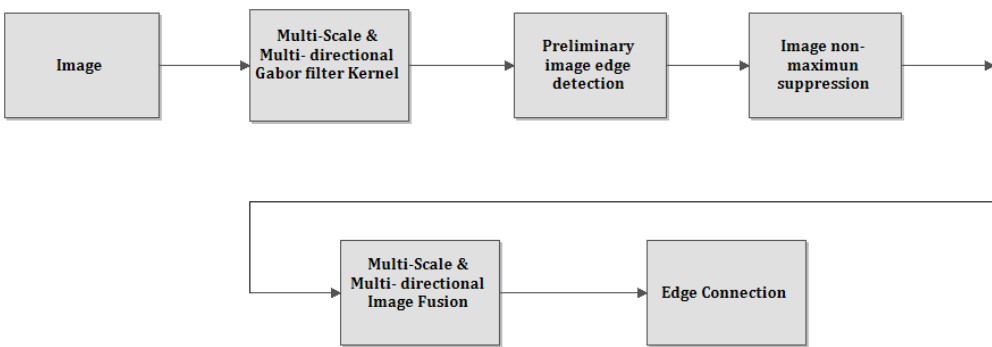

**Figure 3.** Gabor filter-based edge detection process.

The images shown in Figure 2 are selected for edge detection. The edge of the image mainly represents a boundary between the image and the background. Since the Gabor imaginary part is effective for deriving edge information from an image, a series of kernel filters is framed with respective central frequencies and directions as $f = 0.15, 0.3, 0.45$; and

$\theta = 0, pi/4, pi/2, 3pi/4$. As shown in Figure 3, in preliminary image edge detection, the pre-processed image and the created multi-scale and multi-directional Gabor kernel are convoluted to obtain edge patterns in various orientations at different frequencies. The obtained image will provide the edge information for particular data set input images at different scales and directions shown in Figure 4. In the next step of image non-maximum suppression, the comparison is performed between two points that are close to each point of the image with the image obtained in the preliminary edge detection process. If the value that occurs is at the maximum, then it is utilized, or else it becomes zero.

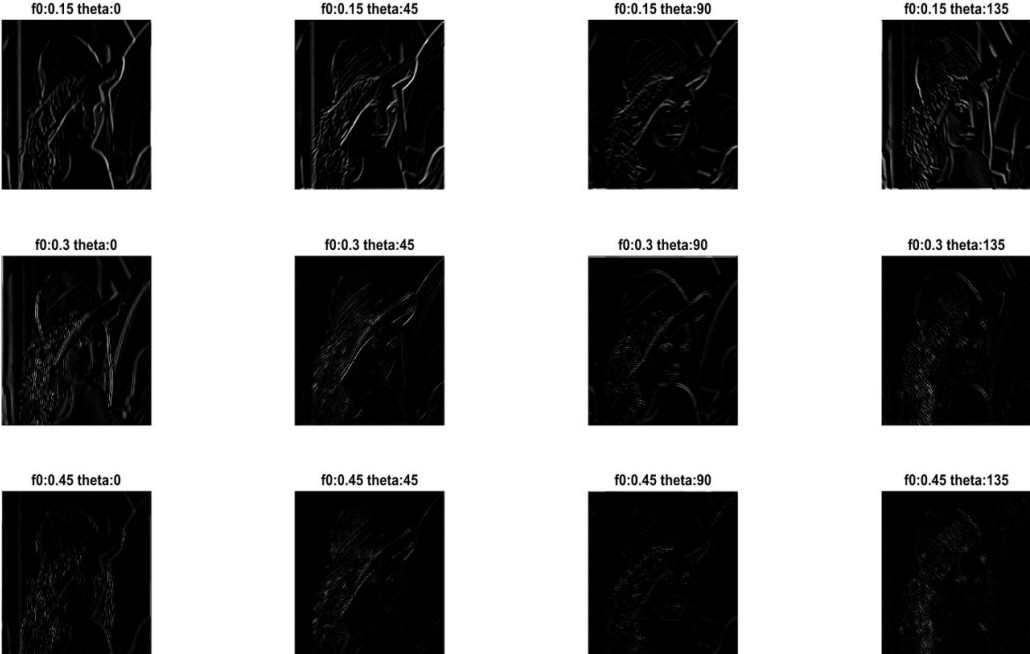

**Figure 4.** Edge information in particular directions and scales for Lena image.

Finally in the image fusion process, the detected images at different scales and different directions contain different edge information. Here, to gain complete edge information, the image needs to be fused. For edge connection, after the fusion process, the edges of the image need to be connected into contours, and the eight neighborhood positions of the fused image are searched for points that can be connected. If it is within the upper and lower threshold range, then consider that it has an edge; if not consider, it has zero edges. The final edge-detected image is shown in Figure 2. The PSNR and MSE metrics of these edge detected images compared to the original images are discussed in Section 5.

## 4. Proposed Architecture of Gabor Filter Hardware

This section describes the hardware architecture of the Gabor filter module. The design of this architecture is performed in line with the study of the Gabor kernel function presented in Section 3. Figure 5 shows the architecture for Gabor filter module where each block represents a part of GF. Based on the statistical expressions and the parameters of the Gabor filter discussed earlier, the hardware architecture is designed in such a manner that each parameter is assigned with a particular value such as $\lambda = 3.5$, $\theta = 0$ or $\pi/4$ or $\pi/2$ or $(3\pi)/4$, $\psi = 0$, $\sigma = 2.8$, $\gamma = 0.3$ and $\pi = 180$. These parameters are determined in a binary weight representation by left-shifting and right-shifting the weights as shown in Figure 6. These are depicted in binary weights in order to know the hardware complexity of the Gabor filter. The entire architecture of the filter module is carried out in a block-by-block process. Figure 6, clearly shows the various parameters determined with a particular value and presented with its binary weight representation.

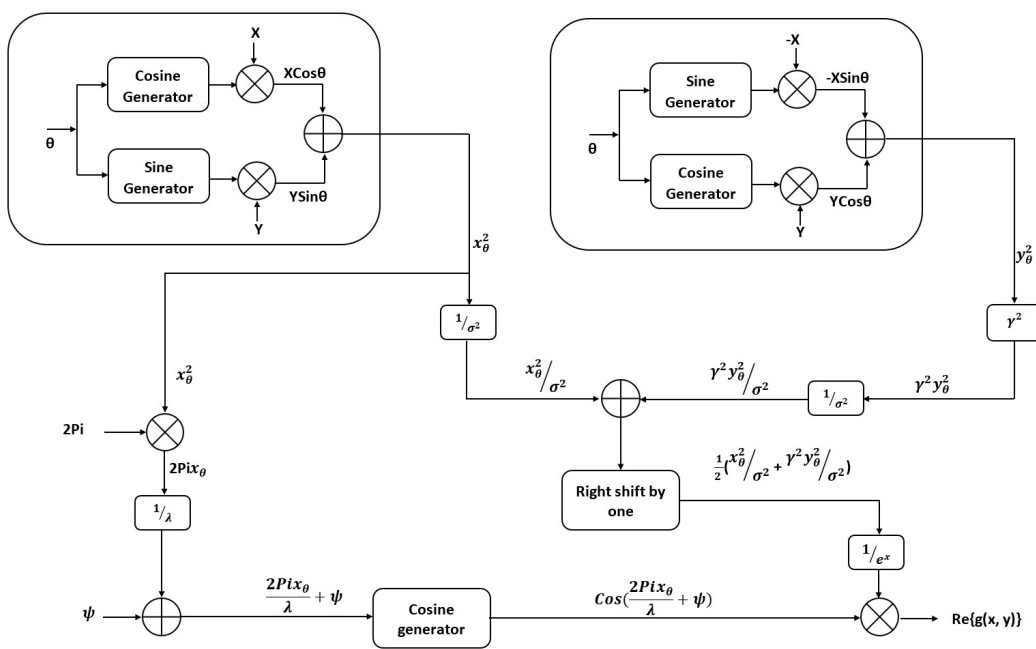

**Figure 5.** Proposed architecture of the Gabor filter model.

| Parameter | Value | Required value | Hardware Realization |
|---|---|---|---|
| Sigma | $\sigma = 2.8$ | $1/\sigma^2 = 0.125$ | 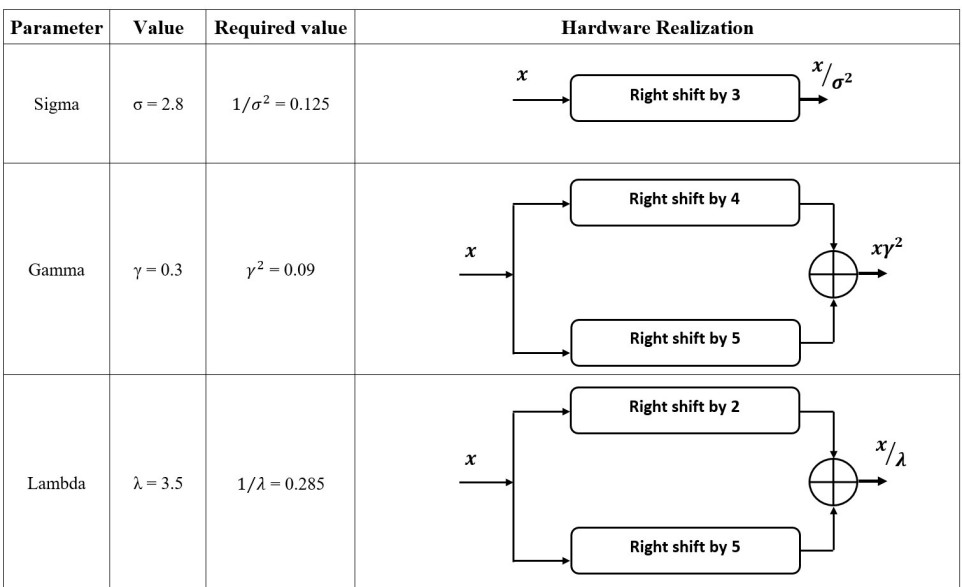 |
| Gamma | $\gamma = 0.3$ | $\gamma^2 = 0.09$ | |
| Lambda | $\lambda = 3.5$ | $1/\lambda = 0.285$ | |

**Figure 6.** Hardware models for realizing the fractional values of parameters involved in the Gabor function.

The real part of the filter as represented in Equation (10) smooths the image while the imaginary part is used for edge detection. In the architecture representation of Equation (4), $[x', y']$ includes the spatial coordinates and $[x, y]$ denotes the coordinates of the input image pixel; '$\theta$' is the angle at particular orientation from 0 to 360 degrees. For better understanding the purpose of the design architecture, $[x', y']$ is denoted as $[x_\theta, y_\theta]$ and their values are given by Equation (4). The $\frac{x_\theta^2}{\sigma^2}$ is obtained by passing $x_\theta^2$ through $\frac{1}{\sigma^2}$ as shown in Equation (14),

$$x_\theta^2 \cdot \frac{1}{\sigma^2} \Rightarrow \frac{x_\theta^2}{\sigma^2} \tag{14}$$

An adder is used to obtain $\frac{x_\theta^2 + \gamma^2 y_\theta^2}{\sigma^2}$ as represented in Equation (15).

$$\frac{x_\theta^2}{\sigma^2} + \frac{\gamma^2 \cdot y_\theta^2}{\sigma^2} \Rightarrow \frac{x_\theta^2 + \gamma^2 y_\theta^2}{\sigma^2} \tag{15}$$

A right shifter is used to obtain the term shown in Equation (16) from the term shown in Equation (15).

$$\frac{1}{2} \cdot \frac{x_\theta^2 + \gamma^2 y_\theta^2}{\sigma^2} \Rightarrow \frac{x_\theta^2 + \gamma^2 y_\theta^2}{2\sigma^2} \tag{16}$$

An exponential module isused to obtain the term shown in Equation (17) from Equation (16).

$$e^{-\frac{1}{2} \cdot \frac{x_\theta^2 + \gamma^2 y_\theta^2}{\sigma^2}} \tag{17}$$

On the other part, $\frac{1}{\lambda}$ is multiplied with $2\pi x_\theta$ and summed with phase($\psi$). A cosine generator is used to obtain $cos(2\pi \frac{x_\theta}{\lambda} + \psi)$ as shown in Equation (18).

$$cos(2\pi \frac{x_\theta}{\lambda} + \psi) \tag{18}$$

The aggregate value comes from both Equations (17) and (18) multiplied to provide a 2D Gabor filter equation for the real part in Equation (10). The same process is carried out for the imaginary part also but by replacing the Cosine generator with a Sine generator, which is used for edge detection. Thus, the output equation occurring for the 2D imaginary part of the Gabor filter is obtained by multiplying Equation (17) along with $sin(2\pi x_\theta / \lambda + \psi)$.

## 5. Experimental Results and Discussions

This section presents the implementation and performance comparisons, among the various methods discussed in the literature. The numerical outputs obtained from the proposed design are validated in the MATLAB environment. Our objective is to detect or extract clean edges. The histograms for the original images for 'Apple', 'Flower' and 'Lena' are shown in Figure 7. Figure 4 shows the edge information for image 'Lena' at various directions and frequencies. Histogram outputs for the images applied with Gabor filter are shown in Figure 7. The histogram calculated for the images produces a graph between intensity levels on the *X*-axis and the number of pixels on the *Y*-axis. The *X*-axis contains all gray levels and the *Y*-axis denotes the number of pixels that have a particular gray-level value.

The Gabor filter is a better feature detection method in which it produces thick edges. This filter provides better results in such cases, such as extracting the edges of facial features of an human beings, animals or any other object. From Figure 2, it is seen that the images of an apple, flower, the statue of liberty, sunflower and lena extract thick edges from the input image. A statistical analysis for the input images that are tested has been achieved in the parameters by changing its orientation, illumination, expression, edge enhancement and background. For every change in parameter value, a change in the output image will occur. Apart from the above-considered input images shown, a few more sample images have been taken to calculate PSNR and MSE to make sure that the proposed method gains better values and obtains high capability in edge detection, i.e., to determine the boundaries of an image, as shown in Figure 8.

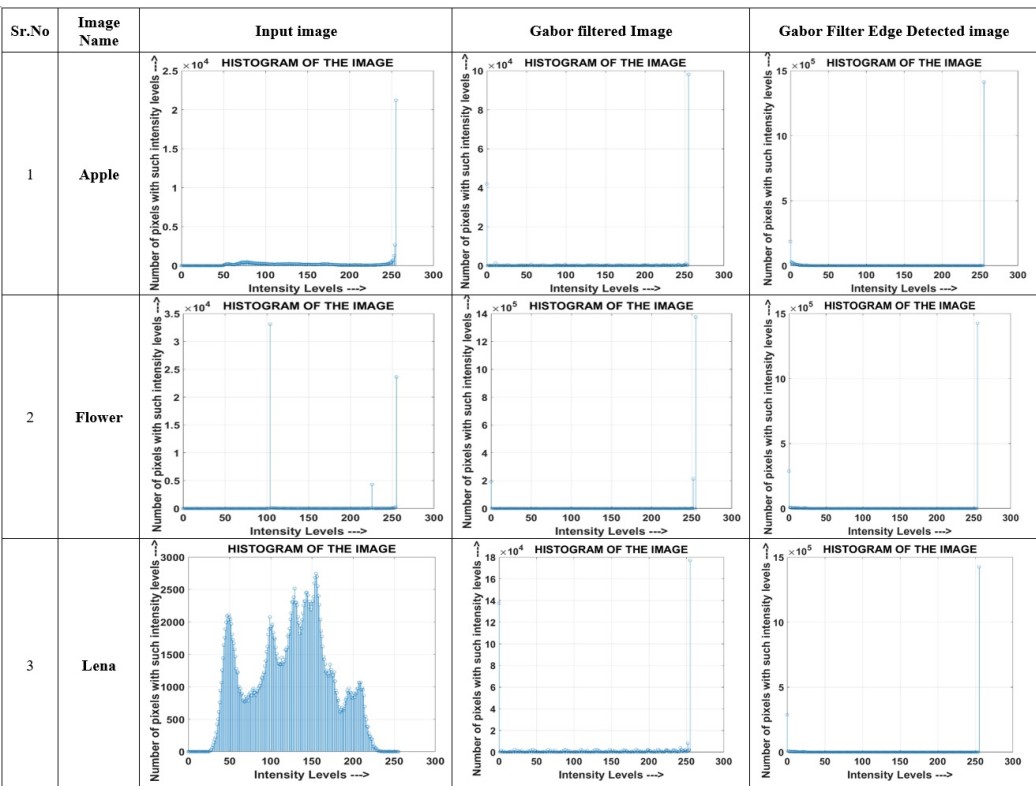

**Figure 7.** Histogram for original image, Gabor filtered image and the image after Edge detection. X label in each histogram: intensity level; Y label in each histogram: number of pixels against the intensity level.

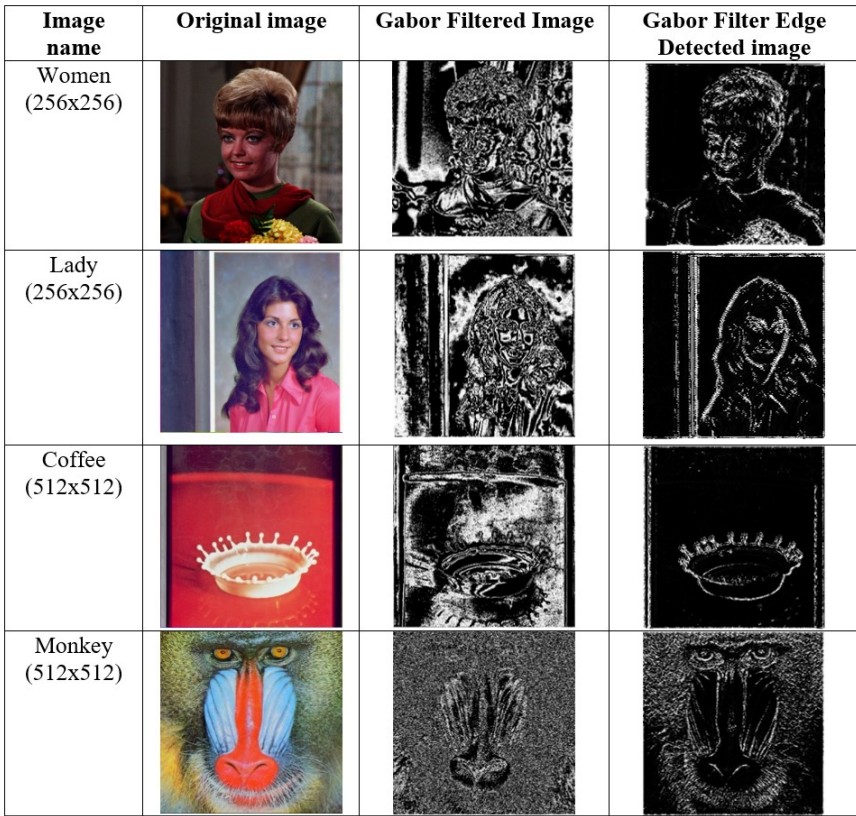

**Figure 8.** Gabor filter and edge detection techniques on various original images.

### 5.1. Performance Comparison and Analysis

In this paper, PSNR and MSE values are taken to compute the execution of the edge detection method. The PSNR determines the peak-SNR between two images. It is measured in decibels (dB). The ratio is measured between the original image and the compressed image or reconstructed image. PSNR represents the peak error. A larger value of PSNR provides a good quality image.

MSE calculates the average squared error between the compressed image and the original image. A lower value of MSE provides better, high-quality images with lower errors. Both Peak-SNR and Mean Squared Error are used to differentiate image quality. MSE is calculated by the formula provided in Equation (19):

$$MSE = \frac{\sum_{M,N}[I1(m,n) - I2(m,n)]^2}{M \times N} \tag{19}$$

where $M$ and $N$ are the rows and columns of the image, $I1$ is the original image and $I2$ is the compressed image. The PSNR ratio is given by the following expression.

$$PSNR = 10log_{10}(\frac{R^2}{MSE}) \tag{20}$$

In Equation (20), $R$ is the maximum pixel value in input data. $R = 255$, when image is 8-bit unsigned integer data type.

The results given in Table 1 are obtained for the images applied with the Gabor filter-based edge detection method. PSNR and MSE results are calculated for each image in Table 1 with the help of Equations (19) and (20). The obtained PSNR and MSE values obtained for the image 'Apple' are 10.4857 and $5.8602 \times 10^3$. The PSNR value for 'Flower' image is 9.2483, and the MSE value is $7.7920 \times 10^3$. The PSNR and MSE values gained for image 'Lena' is 11.8913 and $4.2398 \times 10^3$; for the remaining input images, the output values are given in Table 1.

**Table 1.** Results of PSNR and MSE values for different image datasets applied with the Gabor filter-based edge detection.

| S. No. | Image Name | PSNR (Gabor) | MSE (Gabor) |
|--------|-----------|--------------|-------------|
| 1 | Apple (256 × 256) | 10.4857 | $5.8602 \times 10^3$ |
| 2 | Flower (256 × 256) | 9.2483 | $7.7920 \times 10^3$ |
| 3 | Women (256 × 256) | 10.0816 | $6.4315 \times 10^3$ |
| 4 | Lady (256 × 256) | 9.4466 | $7.4442 \times 10^3$ |
| 5 | Lena (512 × 512) | 11.8913 | $4.2398 \times 10^3$ |
| 6 | Coffee (512 × 512) | 12.5336 | $3.6569 \times 10^3$ |
| 7 | Monkey (512 × 512) | 8.5536 | $9.1437 \times 10^3$ |
| 8 | Sunflower (512 × 512) | 12.3414 | $3.8224 \times 10^3$ |
| 9 | Statue of Liberty (512 × 512) | 12.0976 | $4.0431 \times 10^3$ |

### 5.2. Simulation Results and Comparison Analysis

The hardware architecture of the proposed design is compared with different designs reported in the literature, as shown in Table 2. The following observations were made in this comparison with respect to the proposed design:

1. An operating frequency is at its maximum in FPGA and std_cell implementation [18–20];
2. Performance of edge detection is better in the proposed design compared to other edge operators [29–32];

3. The evaluation of the design in terms of PSNR and MSE is considered with other edge detection methods [30–34].

In Table 2, the proposed work is compared with the relevant existing works in terms of PSNR and MSE measures to assess the quality of the edge detected image. It is important to note that the higher the value of PSNR and lower the value of MSE, the better the image quality. As seen in Table 2, PSNRs of 10.4857, 9.2483, 11.8913, 12.3414 and 12.0976 dB were obtained for the images apple, flower, Lena, sunflower and the statue of liberty and mean square errors of $5.8602 \times 10^3$, $7.7920 \times 10^3$, $4.2398 \times 10^3$, $3.8224 \times 10^3$ and $4.0431 \times 10^3$, respectively, were obtained. Thus, it can be seen that the proposed method has a high value in terms of PSNR and less MSE for all the input images compared with the previous reported designs. The better results in terms of PSNR and MSE for the proposed work are obtained because the respective image is denoised using the proposed Gabor filter before edge detection. To have a fair comparison criterion, PSNR and MSE calculations are performed in the proposed work using similar images considered by all other approaches. The Gabor filter edge detection method has a high capability in detecting edges than other edge detection operators. However, this filtering method does not provide a proper appearance of the image, but it produces unique patterns for different image expressions and produces thick edges as discussed.

**Table 2.** Performance comparison with existing works.

| Ref. | Method | Image Name | PSNR (dB) | | MSE |
|---|---|---|---|---|---|
| Proposed | Gabor Filter + Edge detection + Hardware Accelerator | Apple | 10.4857 | | $5.8602 \times 10^3$ |
| | | Flower | 9.2483 | | $7.7920 \times 10^3$ |
| | | Lena | 11.8913 | | $4.2398 \times 10^3$ |
| | | Sunflower | 12.3414 | | $3.8224 \times 10^3$ |
| | | Statue of Liberty | 12.0976 | | $4.0431 \times 10^3$ |
| [31] | Edge detection algorithm (Canny) | Apple | 2.678 | | $3.54 \times 10^4$ |
| | | Flower | 2.789 | | $3.45 \times 10^4$ |
| | | Lena | 6.206 | | $1.57 \times 10^4$ |
| [29] | Edge detection (Prewitt, Sobel, Laplacian of Gaussian, Canny, Roberts) | Salt Pepper Noise Effected Sun Flower Image | Roberts | 8.6962 | $8.779 \times 10^4$ |
| | | | Sobel | 8.6965 | $8.778 \times 10^4$ |
| | | | Prewitt | 8.6965 | $8.778 \times 10^4$ |
| | | | LOG | 8.7024 | $8.766 \times 10^4$ |
| | | | Canny | 8.7087 | $8.754 \times 10^4$ |
| [32] | Edge Detection (Canny operator, Laplacian of Gaussian, Sobel operator) | Lena | LOG | 5.2217 | $1.953 \times 10^4$ |
| | | | Canny | 5.2161 | $1.956 \times 10^4$ |
| | | | Sobel | 5.2476 | $1.942 \times 10^4$ |
| [30] | Edge detection (Sobel, Prewitt, Canny) | Statue of liberty | Sobel | 5.6365 | $1.775 \times 10^4$ |
| | | | Prewitt | 5.6342 | $1.776 \times 10^4$ |
| | | | Canny | 5.6182 | $1.783 \times 10^4$ |

In Table 3, the method followed in the proposed work is compared with the other benchmark methods in terms of three reference image-based metrics: L2RAT [35], SSIM, and FSIM [36,37]. Here, L2RAT is the ratio of the squared norm of the approximated edge image to the original image. The Structural Similarity Index Metric (SSIM) measures the similarity of edge images with the original images. The algorithm that offers the highest values of SSIM is considered the best one. Feature similarity index metric (FSIM) measures image quality with reference to similarity in features. These metrics of value close to one

are desirable for appraising the given method. From Table 3, it can be observed that the values of L2ART and SSIM are better compared to the other methods since the respective values of the proposed methods are closest to one among all methods. Moreover, it is observed that FSIM for the proposed method showed mixed performance.

**Table 3.** Image quality assessment and comparison.

| Image | Method | L2RAT | SSIM | FSIM |
|---|---|---|---|---|
| Apple | Sobel | $5.64 \times 10^{-7}$ | $2.00 \times 10^{-6}$ | 0.032 |
| | log | $1.49 \times 10^{-6}$ | $4.91 \times 10^{-6}$ | 0.046 |
| | Canny | $1.90 \times 10^{-6}$ | $3.47 \times 10^{-6}$ | 0.823 |
| | Prop. | 0.8696 | 0.779 | 0.788 |
| Lena | Sobel | $2.23 \times 10^{-6}$ | $9.75 \times 10^{-7}$ | 0.286 |
| | log | $3.62 \times 10^{-6}$ | $8.04 \times 10^{-6}$ | 0.754 |
| | Canny | $5.26 \times 10^{-6}$ | $2.31 \times 10^{-7}$ | 0.568 |
| | Prop. | 0.7819 | 0.3992 | 0.383 |
| Sunflower | Sobel | $1.84 \times 10^{-6}$ | $1.19 \times 10^{-6}$ | 0.054 |
| | log | $4.42 \times 10^{-6}$ | $1.13 \times 10^{-5}$ | 0.779 |
| | Canny | $7.98 \times 10^{-6}$ | $3.66 \times 10^{-6}$ | 0.13 |
| | Prop. | 0.6218 | 0.3827 | 0.494 |
| Statue | Sobel | $2.04 \times 10^{-6}$ | $1.35 \times 10^{-6}$ | 0.469 |
| | log | $2.48 \times 10^{-6}$ | $8.61 \times 10^{-6}$ | 0.337 |
| | Canny | $3.75 \times 10^{-6}$ | $9.24 \times 10^{-6}$ | 0.794 |
| | Prop. | 0.7625 | 0.3921 | 0.393 |
| Flower | Sobel | $6.75 \times 10^{-7}$ | $4.97 \times 10^{-7}$ | 0.529 |
| | log | $6.86 \times 10^{-7}$ | $6.61 \times 10^{-7}$ | 0.602 |
| | Canny | $8.77 \times 10^{-7}$ | $3.30 \times 10^{-7}$ | 0.654 |
| | Prop. | 0.891 | 0.7699 | 0.737 |

## 6. Conclusions

The work proposes a design of the image denoising hardware accelerator based on the 2D Gabor filtering method. Gabor-based edge detection is performed to form inner and outer edges of the given input image. The MATLAB model of the proposed hardware accelerator is simulated and performance is assessed in terms of PSNR, MSE, and Histogram calculations. By applying different images to the proposed design, it is observed that the PSNRs of 10.4857/9.2483/11.8913/12.3414/12.0976 dB and MSE of $5.8602 \times 10^3 / 7.7920 \times 10^3 / 4.2398 \times 10^3 / 3.8224 \times 10^3 / 4.0431 \times 10^3$ and so on are obtained for Gabor filter-based edge detection images and compared with the theoretically expected values. Histograms for the same output images from the proposed design and from theoretical simulation are acquired to study the frequency of pixel intensities and compared. It is observed that the proposed design is more competitive in its performance with high PSNR and less MSE values. Thus, the proposed Gabor filter-based edge detection method acquires better feature detection with a thick edge or the boundaries of an image compared to the other edge operators. Image quality assessment (IQA) metrics such as L2AR and SSIM for the proposed method are calculated and compared with the benchmark methods where the proposed method showed better performance. The limitations of the design are as follows: On-chip memories are needed because the input images are buffered when edge detection is performed with hardware accelerators. Further improvements can be conducted by integrating the proposed design into an FPGA board. The proposed model-

based design can be used as a reusable IP for any signal processing or image processing integrated circuit. Moreover, this design can be integrated for applications of hardware accelerators such as bit-level computations, signal processing and image processing. This can be deployed in edge detectors processors or denoising processors after developing a reusable IP (intellectual property).

**Author Contributions:** Data curation, Formal analysis, Investigation, V.D. (Virodhi Dakshayani); Methodology, Conceptualization, Supervision, G.R.L.; Supervision, P.P.; Investigation, Supervision, V.D. (Venkataramana Datti); Investigation, C.K. All authors have read and agreed to the published version of the manuscript.

**Funding:** This research received no external funding.

**Data Availability Statement:** Data is contained within the article.

**Conflicts of Interest:** The authors declare no conflict of interest.

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
