# Peer review of "Design of a Gabor Filter-Based Image Denoising Hardware Model"

_electronics, doi:10.3390/electronics11071063_

Round 1

Reviewer 1 Report

1) It is better to have a concise abstract that clearly states the objective of the study (Gabor filter-based edge detection ) and the algorithmic benefit or specialty of the proposed method.

2) I suggest changing "...  the reported methods where ..."  to " ... the reported methods in terms of the Peak-Signal to Noise Ratio, Mean Square Error, Histogram of the filtered and edge detected image." in the last sentence. Detailed PSNR value and MSE are not relevant in the abstract.   

3) The manuscript does not clearly present what the newly 'proposed method', 'approach in implementation' or 'discovery' in this study are, other than the well-known Gabor filtering method.

4) The notation '*' in equation (1) and (3) was used for the multiplication, but it is usually used for the convolution.

5) The dot notation in equation (2) and (3) seemed to represent the multiplication. The common representation of the multiplication is the omission. I.e., a.b --> ab.  

6) Image sizes in Figure 5 are not scaled in a fixed rate. I cannot understand how the size of input image image differs from that of the gray image; how the size of Gabor-filtered image image is smaller than that of the gray image. 

Author Response

Response for Reviewer-1 Comments

Review Comment

Author Response

It is better to have a concise abstract that clearly states the objective of the study (Gabor filter-based edge detection) and the algorithmic benefit or specialty of the proposed method.

The authors would like to thank the reviewer for the suggestion to improve the quality of the article.

The abstract is updated now to state the use of the proposed architecture and the objective of the work.

I suggest changing "...  the reported methods where ..."  to " ... the reported methods in terms of the Peak-Signal to Noise Ratio, Mean Square Error, Histogram of the filtered and edge detected image." in the last sentence.

The authors are thankful to the reviewers for the constructive suggestion.

Suggested changes are incorporated in the manuscript and highlighted in blue color.

Detailed PSNR value and MSE are not relevant in the abstract.

The sentence stating the values of PSNR  and  MSE are removed from the abstract. However, those are discussed in Section 5 and the conclusion.

The manuscript does not clearly present what the newly 'proposed method', 'approach in implementation' or 'discovery' in this study are, other than the well-known Gabor filtering method.

The proposed method and novelty are addressed in Section 1.2. The approach of the algorithm to architecture mapping is addressed in Section-3.

The notation '*' in equation (1) and (3) was used for the multiplication, but it is usually used for the convolution.

The authors are thankful to the reviewers for identifying the typos.

Suggestions are incorporated in the manuscript and highlighted in blue color.

The dot notation in equation (2) and (3) seemed to represent the multiplication. The common representation of the multiplication is the omission. I.e., a.b --> ab. 

The authors are thankful to the reviewers for identifying the typos.

Suggestions are incorporated in the manuscript and highlighted in blue color.

Image sizes in Figure 5 are not scaled in a fixed rate. I cannot understand how the size of input image image differs from that of the gray image; how the size of Gabor-filtered image image is smaller than that of the gray image.

Thank you for identifying the issue.

All the figure sizes are the same as per the experimental results obtained. Unfortunately, the figure sizes are not properly formatted during latex document setup. Now, all the figures are formatted properly as per the reviewer’s suggestion.

Reviewer 2 Report

Dear authors,

I tried to analyse your paper for some errors, mistakes, missing parts or uncomplete ideas.. I found several issues.

What i am missing in the paper, is the more clear defition of problem you would like to solve.. This is more than welcome....

And also just summary what I need to be in article and what already is inside..

- Abstract contain conclusions with values.. OK.. but can be be better..  Some more exact contribution highlight..  LIMITATIONS ARE MISSING..

- graphs/tables from results with values, X and Y axis labels...OK. tables are OK..    

- Figures are not well visible - e.g. Figure 6 - images.. not visible axes.. labels.. values..

- flow chart of solution... OK.. nice..

- references not only in introduction but in wohle article while contribution is based on them.. OK!

- references to this journal - as to prove closenes of the topic - -- there are No One NOT CORRECT. Need to be added some ... and also from IEEE related Q1 journals.. 

future directions in the end of conclusion.. OK.

reasonable number of references.. ok..

Q1/Q2 journal articles in references.. OK..

So please update as MINOR revisions from me!

Author Response

Response for Reviewer-2 Comments

What i am missing in the paper, is the clearer defition of problem you would like to solve. This is more than welcome....

The authors are thankful to the reviewers for the constructive suggestion.

The suggestion is addressed in the abstract and Section 1.2.

And also just summary what I need to be in the article and what already is inside..

The authors would like to thank the reviewer for their endorsement.

Abstract contain conclusions with values. OK. but can be better.  Some more exact contribution highlight.  LIMITATIONS ARE MISSING..

The authors are thankful to the reviewers for the constructive suggestion.

The abstract is now modified to highlight the contribution of the work. The limitation is presented in the last sentence of the abstract.

graphs/tables from results with values, X and Y axis labels...OK. tables are OK

The authors would like to thank the reviewer for their endorsement.

Figures are not well visible - e.g. Figure 6 - images... not visible axes. labels. values.

Thank you for identifying the issue.

The details of the X & Y axes labels are addressed in the caption of Figure 6 in the manuscript and highlighted in blue.

flow chart of solution. OK. nice.

The authors would like to thank the reviewer for their endorsement.

references not only in introduction but in whole article while contribution is based on them. OK!

The authors would like to thank the reviewer for their endorsement.

references to this journal - as to prove closeness of the topic - -- there are No One NOT CORRECT. Need to be added some ... and also from IEEE related Q1 journals..

The authors are thankful to the reviewers for the constructive suggestion.

Relevant IEEE Q1 journals are added: [1], [7], [20], [25], [26], [4], [14]

future directions in the end of conclusion. OK.

The authors would like to thank the reviewer for their endorsement.

reasonable number of references. ok..

The authors would like to thank the reviewer for their endorsement.

Q1/Q2 journal articles in references. OK..

The authors would like to thank the reviewer for their endorsement.

So please update as MINOR revisions from me!

All suggestions are incorporated in the manuscript and highlighted in blue color.

Reviewer 3 Report

The authors propose the design of a hardware accelerator for the calculation of Gabor filters.

After the review of the paper, I miss an adequate contribution of the work. Despite the comparison with other works in my opinion it is not enough the numbers of table 2 to conclude that the proposed approach is superior to the state-of-the-art works. Please try to explain better the results of the PSNR for the proposed filter and also explain why the other approaches do not use the same measure to be compared to each other.

I found several writing errors, below are some of them. Consider also the review of the use of the articles ”the” and “a”.

with in, image and compared with the, internet of things (IOT), Gabor filter are, that has a capability to, by a HW, sasirekha, detecting edge by, design targets to a, methods those are partitioned into, designed for the Edge detection, of band pass, be smaller or larger size, where kernels with larger size handles, etc.

Author Response

Response for Reviewer-3 Comments

1

I miss an adequate contribution of the work.

The authors are thankful to the reviewers for the constructive comment.

The proposed method and novelty are addressed now in the abstract & Section 1.2. The approach of the algorithm to architecture mapping is addressed in Section-3.

2

Despite the comparison with other works in my opinion it is not enough the numbers of table 2 to conclude that the proposed approach is superior to the state-of-the-art works.  Please try to explain better the results of the PSNR for the proposed filter and also explain why the other approaches do not use the same measure to be compared to each other.

The authors are thankful to the reviewers for the constructive comment.

Mapping the architecture from the Gabor-based image denoising math is the main contribution of this paper. MATLAB models of the proposed architecture are developed and simulated to generate the performance metrics. Further, the performance is compared with the algorithm-level performance reported in the literature. Hence, the performance of the proposed hardware model is close to the performance of the benchmarking algorithms reported in the literature. 

4

I found several writing errors, below are some of them. Consider also the review of the use of the articles ” the” and “a”:

with in, image and compared with the, internet of things (IOT), Gabor filter are, that has a capability to, by a HW, sasirekha, detecting edge by, design targets to a, methods those are partitioned into, designed for the Edge detection, of band pass, be smaller or larger size, where kernels with larger size handles, etc.

The authors would like to thank the reviewer for the suggestion to improve the quality of the article.

Suggestions are incorporated throughout the manuscript and highlighted in blue color.

Round 2

Reviewer 1 Report

The authors revised the manuscript as suggested. I think the revision improves  the manuscript a lot. 

Author Response

Review Response - Manuscript ID: electronics - 1497007 – Revision2

Author’s Response: Many thanks to editors and reviewers for their valuable comments/suggestions to improve this article. The article is revised according to the comments and suggestions given below. The modifications in the manuscript are highlighted in green color.

Response for Reviewer-1 Comments

S. No.

Review Comment

Author Response

1

The authors revised the manuscript as suggested. I think the revision improves the manuscript a lot.

Dear reviewer, thank you for your positive comment regarding my manuscript.  

Reviewer 3 Report

The authors present a new version of an article already summited for revision. After a review of the new version, sorry but I couldn't find an important improvement in comparison with the first version, so my observations are the same as in the first submission. Some writing errors are corrected but the english languaje and style must be improved.

Author Response

Response for Reviewer-3 Comments

1

I miss an adequate contribution of the work.

Dear reviewer, thank you very much for your constructive comments for the fruitful improvement of our manuscript.

As per your valuable suggestion, the main contribution of the proposed work is now clearly explained in the revised manuscript in section 1.2.  In connection to this, to clarify our contribution, the approach of mapping the algorithm to architecture is also addressed in section 3. Moreover, we have completely modified the abstract in the revised manuscript for the better readability.

2

Despite the comparison with other works in my opinion it is not enough the numbers of table 2 to conclude that the proposed approach is superior to the state-of-the-art works.  Please try to explain better the results of the PSNR for the proposed filter and also explain why the other approaches do not use the same measure to be compared to each other.

Dear reviewer, thank you for your valuable comment.

In Table 2, the proposed work is compared with the relevant existing works in terms of PSNR and MSE measures to assess the quality of the edge detected image.  

The better results in terms of PSNR and MSE for the proposed work are obtained because the respective image is denoised using the proposed Gabor filter before edge detection.

To have a fair comparison criterion, the PSNR and MSE calculations are done in the proposed work using the similar images considered by all other approaches.  

Moreover, other approaches are also used the PSNR & MSE measures in their comparison analysis.

From Table 2, it is clearly shown that the PSNR and MSE values of our proposed work are better (PSNR higher & MSE is lower) compared to the other existing works for the different kind of images like Apple, Lena etc.

The above clarifications are included in the manuscript and highlighted in green.

3

I found several writing errors, below are some of them. Consider also the review of the use of the articles ” the” and “a”:

with in, image and compared with the, internet of things (IOT), Gabor filter are, that has a capability to, by a HW, sasirekha, detecting edge by, design targets to a, methods those are partitioned into, designed for the Edge detection, of band pass, be smaller or larger size, where kernels with larger size handles, etc.

The authors would like to thank the reviewer for the suggestion to improve the quality of the article.

As per your valuable suggestion, the entire revised manuscript is thoroughly checked typographical errors. Moreover, your suggestions regarding the typo errors are also incorporated in the revised manuscript.

Round 3

Reviewer 3 Report

In my opinion, and sorry for saying that, the new version of the article has only a small improvement in comparison with the original one. So I don't see new evidence to reconsider the last recommendation. Please note that in your new text in green there are new writing errors. I would recommend reformulating the research objectives of your investigation and then defining the tests that you need to prove that your proposed architecture is superior or at least similar to the ones in the state of the art. Please consider different quality measures (not only PSNR and MSE) to assess the produced images.

Author Response

Review Response - Manuscript ID: electronics - 1497007 – Revision3

Author’s Response: Many thanks to editors and reviewers for their valuable comments/suggestions to improve this article. The article is revised according to the comments and suggestions given below. The modifications in the manuscript are highlighted in BLUE color.

Response for Reviewer-3 Comments

S. No.

Review Comment

Author Response

1

Please note that in your new text in green there are new writing errors.

I would recommend reformulating the research objectives of your investigation and then defining the tests that you need to prove that your proposed architecture is superior or at least similar to the ones in the state of the art. Please consider different quality measures (not only PSNR and MSE) to assess the produced images.

The authors would like to thank the reviewer for the suggestion to improve the quality of the article.

The writing errors are rectified.

In Table 3, the method followed in the proposed work is compared with the other benchmark methods in terms of three image quality assessment metrics viz. L2RAT, SSIM, and FSIM. Respective table (Table.3) and the discussion are added in Section 5.2. It is observed that the proposed method showed better performance.
